# Effects of PCSK9 inhibitors on coronary microcirculation, inflammation and cardiac function in patients with CHD after PCI: a protocol for systematic review and meta-analysis

Xuejiao Ye [iD],[1] Shihan Wang,[2] Xiao'an Liu,[3] Qian Wu,[1] Yanfei Lv,[4] Qianyu Lv,[1] Junjia Li,[1] Lanlan Li,[1] Yingtian Yang [iD] [5]

[1]Guang'anmen Hospital, China Academy of Chinese Medical Sciences, Beijing, China
[2]Department of Cardiology, China Academy of Traditional Chinese Medicine Guang'anmen Hospital, Beijing, China
[3]Capital Medical University, Beijing, China
[4]Shanghai Qianhe Technology Co LTD, Shanghai, China
[5]Beijing University of Chinese Medicine, Beijing, China

**Correspondence to**
Dr Shihan Wang;
wangshihan91@126.com

## ABSTRACT

**Introduction** Coronary heart disease (CHD) is one of the common cardiovascular diseases that seriously jeopardise human health, and endothelial inflammation and dyslipidaemia are the initiating links leading to its occurrence. Percutaneous coronary intervention (PCI) is one of the most effective surgical treatments for CHD with narrowed or blocked blood vessels, which can quickly unblock the blocked vessels and restore coronary blood supply. However, most patients may experience coronary microcirculation disorders (CMDs) and decreased cardiac function after PCI treatment, which directly affects the efficacy of PCI and the prognosis of patients. Preprotein converting enzyme subtilisin/Kexin 9 (PCSK9) inhibitors are novel pleiotropy lipid-lowering drug with dual anti-inflammation and lipid-lowering effects, and represent a new clinical pathway for rapid correction of dyslipidaemia. Therefore, we designed this protocol to systematically evaluate the effects of PCSK9 inhibitors on coronary microcirculation and cardiac function in patients with CHD after PCI, and to provide high-quality evidence-based evidence for the clinical application of PCSK9 inhibitors.

**Methods and analysis** This protocol is reported strictly in accordance with the 2020 Preferred Reporting Items for Systematic Reviews and Meta-analyses Protocols Guidelines. We will search PubMed, EMBASE, Web of Science and three Chinese databases (CNKI, Wanfang and VIP database) according to preset search strategies, without language and publication data restrictions. We will work with manual retrieval to screen references that have been included in the literature. Google Scholar will be used to search for grey literature. The final included literature must meet the established inclusion criteria. Titles, abstracts and full text will be extracted independently by two reviewers, and disagreements will be resolved through discussion or the involvement of a third reviewer. Extracted data will be analysed using Review Manager V.5.3. The Cochrane Risk of Bias Tool will be used to evaluate the risk of bias. Publication bias will be assessed by funnel plots. Heterogeneity will be assessed by $I^2$ test and subgroup analyses will be used to further investigate potential sources of heterogeneity. The quality of the literature will be assessed by GRADE score. This protocol will start in January 2026 and end in December 2030.

**Ethics and dissemination** This study is a systematic review of published literature data and no special ethical approval was required.

**PROSPERO registration number** CRD42022346189.

## STRENGTHS AND LIMITATIONS OF THIS STUDY

⇒ To the best of our knowledge, this protocol is an updated systematic review and meta-analysis designed to assess the effects of PCSK9 inhibitors on coronary microcirculation, inflammation and cardiac function in patients with coronary heart disease after percutaneous coronary intervention.

⇒ Compared with the previous studies, we considered a wider range of outcome indicators, including laboratory indicators and ultrasound indicators.

⇒ Subgroup and sensitivity analyses will be used to explore potential heterogeneity.

⇒ The limited number of studies that meet our inclusion criteria is a limitation of this study.

## INTRODUCTION

Coronary heart disease (CHD) is a cardiovascular disease caused by coronary atherosclerosis, which causes narrowing or occlusion of the lumen of the blood vessels, resulting in myocardial ischaemia, hypoxia or necrosis.[1] Percutaneous coronary intervention (PCI), coronary artery bypass grafting (CABG) and lipid management are three effective treatments for CHD. And PCI, in particular, is the mainstay of treatment for coronary artery multibranch vasculopathy in the clinic. It is widely used in clinic practice.[2] It can effectively unblock the blocked vessels and realise the reconstruction of coronary blood flow.[3] Accompanying obvious benefits, there are also certain risks. Invasive procedures such as balloon dilation during PCI can damage

vascular endothelium, induce an inflammation in endothelial cells, and promote microthrombosis, leading to coronary microcirculatory disorders (CMDs) and myocardial ischaemia-reperfusion injury (MIRI), which significantly increase the risk of major adverse cardiovascular events (MACEs) in patients during the perioperative period.[4 5]

The coronary microcirculation consists of microvessels with diameters of <500 μm (microarterioles, capillaries and microvessels), which collectively participate in maintaining the normal physiological function of cardiomyocytes and the stability of the coronary microcirculation. CMD refers to the structural and functional abnormalities of the coronary vasculature, and endovascular obstruction and extravascular compression are the basis for the occurrence of CMD.[6] When coronary angiography is normal and there are no other diseases, CMD is usually the main reason for the occurrence of myocardial ischaemia.[7] The ISCHEMIA study published at the 2019 American Heart Association meeting showed that the risk of MACEs in patients with CHD with moderate myocardial ischaemia is not significantly reduced after completion of revascularisation, as expected.[8] Moreover, in conjunction with clinical practice, we found that some patients with CHD did not experience improvement in angina symptoms after PCI, nor did the myocardium recover to full reperfusion levels. All of these studies suggest that potential CMD may be responsible for influencing the prognosis of patients with CHD.[9] Due to the fact that coronary angiography can only detect 5% of coronary branches, and the remaining 95% of coronary microcirculation cannot be clearly visualised, CMD is easily misdiagnosed and greatly increases the risk of MACEs.[10] Therefore, clarifying the underlying mechanisms of microvascular injury after PCI is of great clinical significance in identifying therapeutic targets for CMD.

TIMI classification is one of the indicators to evaluate the recovery of coronary artery flow reperfusion after PCI. It has been found that the increase in serum high-sensitivity C reactive protein (hs-CRP) concentration after PCI is negatively correlated with the coronary TIMI flow classification. This suggests that inflammation may lead to a decrease in TIMI classification and poor recovery of coronary blood flow.[11] In addition, the extracellular matrix (ECM) is important for maintaining the structural stability of cardiomyocytes. Matrix metalloproteinases (MMPs) and tissue inhibitor of metalloproteinase (TIMP) play a key role in the maintenance of ECM stability.[12 13] Matrix metalloproteinases-9 (MMP-9) is an inflammation marker that breaks down collagen in the plaque fibrous cap, causing the fibrous cap of atheromatous plaques to become thinner and easier to rupture, which in turn destabilises the plaque and induces thrombosis.[14] Studies have confirmed that the activity of MMP-9 in unstable plaques is 3–5 times higher than that in stable plaques. This suggests that the higher the activity of MMP-9, the stronger the degradation of ECM and the easier the plaques rupture.[15] TIMP is an inhibitor of MMPs, which is expressed by smooth muscle cells under

physiological conditions. TIMP plays an important role in preventing ventricular remodelling after myocardial ischaemia by inhibiting the biological activity of MMPs and reducing the degradation of the ECM. The balance between MMP-9 and TIMP jointly maintains the stability of atherosclerotic plaques. When MMP-9 levels decrease and TIMP levels increase, ECM degradation decreases and plaques become more stable. Conversely, ECM degradation increases, and plaques become destabilised and more prone to rupture, increasing the risk of acute coronary events.[16] Based on this, we hypothesised that inflammation factors such as MMP-9 may be biomarkers for assessing the prognosis of patients with CHD and may become new therapeutic targets.

Currently, several clinical studies have evaluated the cardioprotective effects of PCSK9 inhibitors after PCI, but there is no relevant systematic literature review on this topic. Therefore, we designed this protocol to systematically summarise the effects of PCSK9 inhibitors on coronary microcirculation and cardiac function in patients with CHD after PCI, and to provide high-quality evidence-based medical evidence for the clinical application of PCSK9 inhibitors.

### Objective
1. Main efficacy indicators: TIMI flow classification of coronary arteries, myocardial injury markers (cardiac troponin (cTn), creatine kinase isoenzyme (CK-MB), myoglobin (Myo)), N-terminal pro-brain natriuretic peptide levels (NT-pro-BNP) and D-dimer (D-D) before and after PCI.
2. Secondary efficacy indicators: Serum inflammation markers (CRP/hs-CRP, interleukin(IL-6), tumour necrosis factor-α (TNF-α), matrix metalloproteinase-9(MMP-9)), cardiac function indicators (left ventricular ejection fraction (LVEF), fractional shortening (FS)), blood lipid indicators (low-density lipoprotein cholesterol (LDL-C), total cholesterol (TC), triglyceride (TG), high-density lipoprotein cholesterol (HDL-C)).
3. Safety indicators: The incidence of postoperative complications (puncture point haematoma, gingival bleeding, haematochezia, urinary blood, etc), MIRI (reperfusion arrhythmia (RA), myocardial stunning, no-reflow and lethal reperfusion injury, etc), MACEs (incidence of recurrent myocardial infarction, heart failure, cardiogenic shock and cardiogenic death within 6 months after PCI.[17–19]

### METHODS AND ANALYSIS
#### Protocol registration
This protocol is reported following the 2020 Preferred Reporting Items for Systematic Review and Meta-analysis Protocols statement (PRISMA-P)[20] and has been registered on the International Prospective Register of Systematic Reviews (PROSPERO) network. Registration number is CRD4202346189.

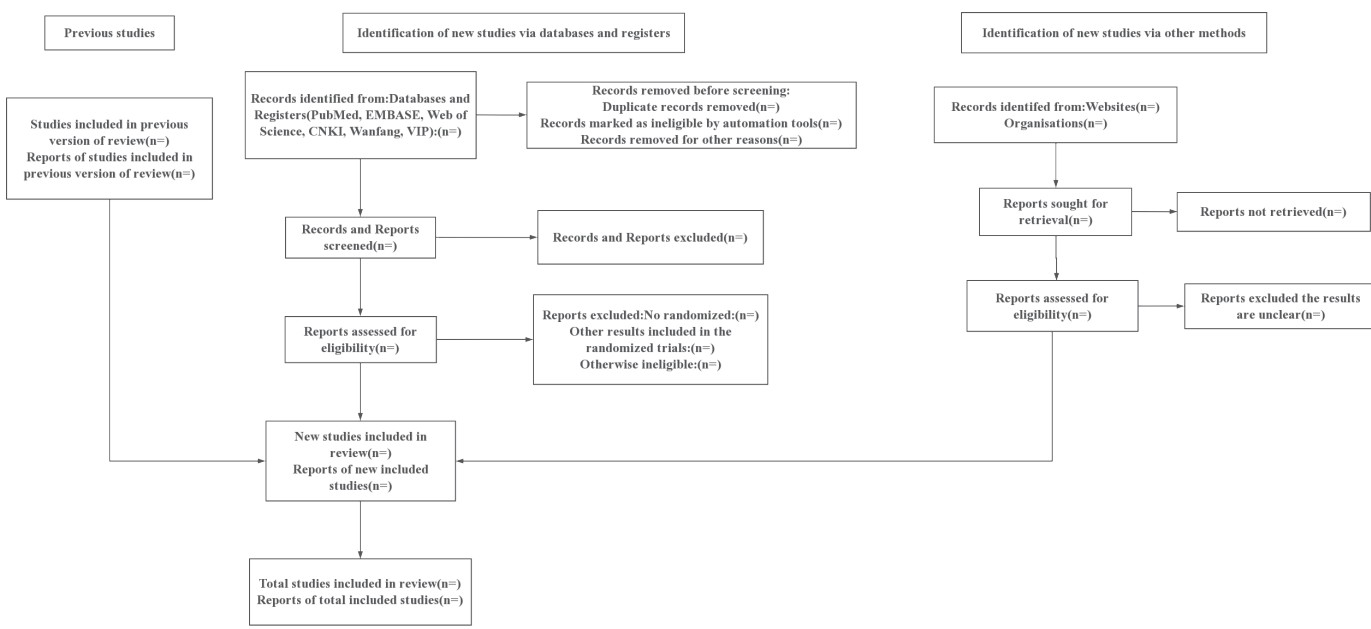

**Figure 1** Search strategy for PubMed.

### Date sources and search strategies

For this article, we will search PubMed, EMBASE, Web of Science and three Chinese databases (CNKI, Wanfang, VIP database). Google Scholar will be used to search for grey literature. We will work with manual retrieval to screen references that have been included in the literature.

Search terms are combinations of free and subject terms, including PCI, PCSK9 inhibitors, inflammation, CHD, etc. Search terms are connected by Boolean operators. We will provide a detailed search strategy for PubMed as shown in figure 1. For other databases, these terms will be modified as needed, and detailed search strategy as shown in online supplemental file 1. There are no language or data restrictions. Duplicates will be automatically removed by EndNote (V.X9, Clarivate Analytics), and the remaining duplicates will be removed manually by comparing authors, titles and publication dates. All literature that meets the inclusion criteria will be further screened by reading the full text to extract the required data, and excluded literature will be required to document the reasons. This process will be conducted independently by two reviewers and any issues and disagreements will be resolved through mutual discussion or discussion with a third reviewer. In accordance with the PRISMA-P guidelines, we will summarise the entire study screening process in a flow chart.

### PCSK9 inhibitors [MeSH]
1. (proprotein convertase subtilisin/kexin type 9 monoclonal antibody) OR (PCSK9 inhibitors) OR (evolocumab) OR (alirocumab) OR (Praluent) OR (Repatha) [Title/Abstract]

### Inflammation [MeSH]
2. (Inflammation) OR (Inflammatory response) OR (Inflammation reaction) [Title/Abstract]

### Randomized controlled trials [MeSH]
3. (RCT) OR (RCTs) OR (Randomized controlled trials) OR (Randomized controlled trial) [Title/Abstract]

### Coronary heart disease [MeSH]
4. (Coronary artery disease) OR (Arteriosclerotic heart disease) OR (CAD) OR (CHD) OR (acute coronary syndrome) OR (chronic coronary syndrome) OR (ACS) OR (CCS) [Title/Abstract]
5. 1 and 2 and 3 and 4

### Eligibility criteria for study selection
#### Study design
Randomised controlled trials (RCTs) comparing the effects of PCSK9 inhibitors on coronary microcirculation, inflammation factors and cardiac function in patients with CHD after PCI will be included without time and language restrictions. Non-RCTs, observational studies, animal experiments, case reports, reviews and conference abstracts will be excluded. This protocol will begin in January 2026 and end in December 2030.

#### Participants
*Inclusion criteria*
1. Male or female patients who meet the diagnostic criteria for CHD (definition criteria of CHD: acute coronary syndrome and chronic coronary syndrome),[21] aged 18–65.[22–24]
2. PCI completed within 72 hours.
3. Patients who are able to tolerate and willing to receive PCSK9 inhibitors or statins or ezetimibe lipid-lowering treatment.

*Exclusion criteria*
1. Intolerant to statins, ezetimibe or PCSK9 inhibitors.
2. Severe liver and kidney dysfunction.
3. Pregnant or lactating women.

4. Malignant tumour or blood system disease.
5. History of severe allergic reactions to lipid-lowering drugs.
6. Unsuitable participants for any other reasons, as judged by the reviewers.

### Types of interventions

Any RCTs using PCSK9 inhibitors before PCI will be included. PCSK9 inhibitors include evolocumab or alirocumab at a dose of 140 mg/2 weeks or 420 mg/4 weeks. Interventions included but not limited to the following:
1. PCSK9 inhibitors versus placebo.
2. PCSK9 inhibitors versus statins.
3. PCSK9 inhibitors+statins versus statins.
4. PCSK9 inhibitors versus statins+ezetimibe.
5. PCSK9 inhibitors+statins versus statins+ezetimibe.
6. PCSK9 inhibitors+statins versus ezetimibe+statins.

### Types of outcome measures

Eligible studies should report at least one of the following outcomes.
1. Main efficacy indicators: TIMI flow classification of coronary arteries, myocardial injury markers (cTn, CK-MB, Myo), NT-pro-BNP and D-D before and after PCI.
2. Secondary efficacy indicators: Serum inflammation markers (CRP/hs-CRP, IL-6, TNF-α, MMP-9), cardiac function indicators (LVEF, FS), blood lipid indicators (LDL-C, TC, TG, HDL-C).
3. Safety indicators: The incidence of postoperative complications (puncture point haematoma, gingival bleeding, haematochezia, urinary blood, etc), MIRI (RA, myocardial stunning, no-reflow and lethal reperfusion injury, etc), MACEs (incidence of recurrent myocardial infarction, heart failure, cardiogenic shock and cardiogenic death within 6 months after PCI).[17–19]

## Data collection and analysis
### Date extraction and management

After screening the literature according to the inclusion and exclusion criteria, we will perform data extraction using a self-made Microsoft Excel data extraction form, which is presented in online supplemental file 2. This process will be performed independently by two reviewers, and any disagreements, if any, will be resolved through mutual discussion or discussion with a third reviewer. The extracted data are presented below:
1. Study identification: title, first author's name, year of publication.
2. Study characteristics: study design (RCTs), duration of follow-up, number of cases.
3. Study population: sample size, mean or median age, gender composition ratio, proportion of patients with diabetes and hypertension, basic disease, basic medication.
4. Interventions: type, dose and frequency of medication, course of treatment.

5. Main efficacy indicators: TIMI flow classification of coronary arteries, myocardial injury markers (c Tn, CK-MB, Myo), NT-pro BNP and D-D before and after PCI.
6. Secondary efficacy indicators: serum inflammation indicators (CRP/hs-CRP, IL-6, TNF-α, MMP-9), cardiac function indicators (LVEF, FS), blood lipid indicators (LDL-C, TC, TG, HDL-C).
7. Safety indicators: the incidence of postoperative complications (puncture point haematoma, gingival bleeding, haematochezia, urinary blood, etc), MIRI (RA, myocardial stunning, no-reflow and lethal reperfusion injury, etc), MACEs (incidence of recurrent myocardial infarction, heart failure, cardiogenic shock and cardiogenic death within 6 months after PCI).[17–19]

### Data synthesis and analysis

We will apply Review Manager V.5.3 provided by the Cochrane Collaboration (http://www-cochrane-org.vpn1.hactcm.edu.cn/) to analyse the obtained data.[25] For continuous variables, mean value and 95% CIs will be used for data analysis. For dichotomous variables, OR and 95%CI will be applied. A $p \geq 0.05$ indicates no statistical significance, and $p < 0.05$ indicates statistical significance. The heterogeneity of the literature will be evaluated by the $I^2$ test. If $p \geq 0.1$ and $I^2 \leq 50\%$, there is no statistical heterogeneity, and the fixed effect model will be used. If $p < 0.1$ and $I^2 > 50\%$, indicates statistical heterogeneity exists, and random effect model will be used.

### Dealing with missing data

For incomplete or missing data, we will contact the corresponding author to obtain it. Otherwise, relevant literature will be excluded.

### Risk assessment of included literature

The Cochrane Collaboration's risk of bias assessment tool (http://www-cochrane-org.vpn1.hactcm.edu.cn/) will be applied to independently assess the bias risk of each included literature.[26] We will assess the risk of bias from six domains: performance bias (blinding of participants and personnel), selection bias (random sequence generation, allocation concealment), detection bias (blinding of outcome assessment), attrition bias (incomplete outcome data), reporting bias (selective reporting) and other potential biases. The quality of assessment is divided into three grades: low risk of bias, some concerns and high risk of bias. For duplicate literature, we will select only the original version. Any doubts and disagreements will be discussed and resolved by two reviewers, and if necessary, we will seek the help of a third reviewer. Review Manager V.5.3 will be used to draw charts for risk assessment.

### Publication of bias

If more than 10 literature were included, we will use the funnel plot to visually assess publication bias. The symmetry of the two sides of the funnel plot indicates the absence of publication bias, while asymmetry of the distribution of the two sides indicates the possible existence of publication bias. If the number of literature was less

than 10, Egger's test will be used for statistical analysis. A p<0.05 indicates the existence of publication bias, and p≥0.05 indicates the absence of publication bias.

## Data analysis

### Assessment of heterogeneity

We will use the $I^2$ test to evaluate the heterogeneity of the included literature, with $I^2 \leq 50\%$ indicating low heterogeneity and $I^2 > 50\%$ indicating high heterogeneity.

### Subgroup analysis and the sources of heterogeneity

When heterogeneity is evident, we will conduct subgroup analysis to investigate the source of heterogeneity:
1. Age.
2. Race.
3. Proportion of patients with hypertension or diabetes.
4. Types and doses of interventions.

### Sensitivity analyses

We will use a one by one exclusion method for sensitivity analysis to evaluate the robustness of the study results. Sensitivity analysis needs to consider various uncertainties, such as small sample size, missing data and methodological quality. After excluding low-quality literature, we need to repeat the analysis.

### Quality evaluation of included literature

We will use The Grading of Recommendations Assessment, Development and Evaluation (GRADE) score[27] to evaluate the quality of included literature, and each literature will be stratified according to high evel of evidence, moderate level of evidence, low level of evidence and very low-level evidence.

## Patient and public involvement

Since this study is a secondary study based on other studies, there will be no direct patient or public involvement in this study.

## DISCUSSION

PCI can effectively unblock the blocked vessels, rapidly restore coronary blood flow and improve myocardial blood supply. However, invasive procedures such as stenting, balloon dilation and myocardial ischaemia reperfusion during PCI can damage the endothelium by inducing platelet aggregation, microvascular spasm, inflammation and myocardial interstitial oedema, among other mechanisms.[6 28] MIRI leads to the overproduction of reactive oxygen species (ROS), which induces oxidative stress, inflammation and endothelial dysfunction, which are the key pathogenic mechanisms that lead to the development of CMD.[29] Bruins *et al*[30] proved the existence of inflammation by measuring indicators such as CRP and IL-6 in the early postoperative period. Baktashian *et al*[31] found that an increase in the serum inflammation factor, hs-CRP, was strongly related with the occurrence of in-stent restenosis. Yang *et al*[32] found that hs-CRP level was positively correlated with the coefficient

of resistance (IMR) of coronary microcirculation after PCI. Recent studies have further confirmed that elevated levels of inflammation factor CRP can directly damage the diastolic function of the vascular endothelium and stimulate the secretion of MMPs and other plaque rupture-promoting substances, thereby interfering with coronary perfusion.[33] All of these studies suggest a potential link between the inflammation and CMD after PCI. Inflammation induces endothelial cell activation and increases ROS production, indirectly promoting the activation and adhesion of platelet.[34] In addition, inflammation and myocardial ischaemia reperfusion can cause vascular endothelial cell injury and increase the permeability of the microvascular walls, which in turn leads to oedema of myocardial tissues, compression of coronary microvessels and increased microcirculation resistance.[35] Endothelial cell injury also activates the body's coagulation mechanism, induces platelet adhesion and aggregation at the site of endothelial injury, and promotes thrombus formation, further aggravating CMD and increasing the risk of in-stent restenosis events.[36]

Dyslipidaemia is not only an important independent risk factor for CHD, but also an important initiator of coronary atheromatous plaque inflammation. It can affect coronary microcirculation through various mechanisms.[37] Large lipid accumulations in the vascular lumen can induce inflammation, endothelial cell injury, ECM remodelling, platelet activation, and thrombosis, which can directly lead to coronary artery stenosis or occlusion.[38] In addition, higher lipid levels can stimulate the vascular endothelium and induce apoptosis of endothelial cell, which ultimately results in coronary atheromatous plaque development. A prospective cohort study[39] found that patients with CMD had high serum TC and LDL-C levels, indicating that hyperlipidaemia may contribute to microvascular injury and CMD. Serum LDL-C level was positively associated with atherosclerosis plaque formation and the occurrence of MACEs.[40] For every 1 mmol/L reduction in LDL-C, there is a 22% reduction in MACEs, a 10% reduction in all-cause mortality and a 20% reduction in the risk of CHD mortality.[41] Effective control of lipids, therefore, is critical to attenuating inflammation and preventing the development of MACEs after PCI.[42] Intensive lipid-lowering therapy, especially lowering serum LDL-C, is important for improving the prognosis of patients with CHD.[43]

Anti-inflammation and lipid management are important interventions for patients with CHD after PCI, and are key to ensuring smooth coronary microcirculation and preventing in-stent thrombosis. Statins are the cornerstone of the treatment of CHD. Their dual lipid-lowering and anti-inflammation effects are key to their positive effect on coronary atherosclerosis.[44] However, due to the existence of 'the 6% effect',[45] doubling the dose of statins can only reduce LDL-C level by a further 6%. adverse effects, such as rhabdomyolysis and hepatic dysfunction, limit the clinical application of statins.[46] As a secreted serine

protease, PCSK9 is mainly synthesised and secreted by the liver. It can promote plaque inflammation by interfering with circulating lipid levels or increasing monocyte infiltration and differentiation.[47–49] PCSK9 inhibitors are a new type of non-statin lipid-lowering drugs. On the one hand, it exerts lipid-lowering and plaque stabilising effects by decreasing the degradation of LDL-R to increase the clearance of LDL-R. On the other hand, it exerts anti-inflammation effects by reducing LDL-C levels and LDL-R mediated platelet activation.[50 51] Both can reduce the risk of MACEs in patients with CHD, and both are safe and well tolerated by the patients.[52–54] In addition, PCSK9 inhibitors have a synergistic effect with statins. ODYSSEY-LONG TERM trial showed that the combination of PCSK9 inhibitors and statins could reduce the risk of MACEs by about 50% in patients with high-risk cardiovascular diseases, which provides a theoretical basis for the safety of clinical application of PCSK9 inhibitors.[55 56] Based on this, we believe that the powerful lipid-lowering and potential anti-inflammation effects of PCSK9 inhibitors could promote the improvement of coronary microcirculation and cardiac function after PCI. This conclusion remains to be confirmed by numerous animal and clinical studies.[57]

### Ethics and dissemination

This study is a systematic review of published literature data and no special ethical approval was required.

**Contributors** XY and SW designed the study. QW, QL and JL created the tables and figures. XY and YL wrote the initial draft of the manuscript. YY, XL and LL revised the manuscript. All authors contributed to the article and approved the final version of the manuscript.

**Funding** This research is supported by Beijing Municipal Science & Technology Commission (Z191100006619025).

**Competing interests** None declared.

**Patient and public involvement** Patients and/or the public were not involved in the design, or conduct, or reporting, or dissemination plans of this research.

**Patient consent for publication** Not applicable.

**Provenance and peer review** Not commissioned; externally peer reviewed.

**ORCID iDs**
Xuejiao Ye http://orcid.org/0009-0001-5944-6918
Yingtian Yang http://orcid.org/0000-0003-1561-7217

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
