## [Reviewer comments · BMJ Open]

ARTICLE DETAILS

TITLE (PROVISIONAL)	Effects of PCSK9 inhibitors on coronary microcirculation, inflammation and cardiac function in patients with CHD after PCI: A protocol for systematic review and meta-analysis.
AUTHORS	YE, Xuejiao; Wang, Shihan; LIU, Xiao'an; Wu, Qian; LV, Yanfei; Lv, Qianyu; Li, Junjia; Li, Lanlan; Yang, Yingtian

VERSION 1 – REVIEW

REVIEWER	Gresele, Paolo University of Perugia, Division of Internal and Cardiovascular Medicine, Department of Medicine
REVIEW RETURNED	03-May-2023

GENERAL COMMENTS	This is a protocol for a metanalysis role of PCSK9 inhibitors in patients with coronary heart disease(CHD), with special attention on the effects on coronary microcirculation, inflammatory biomarkers and cardiac function. There are several major limitations of the article, besides the quality of writing which definitely requires native speaker revision, including: - Methods-Table 1 and search strategy: it is not clear where the authors refer to CHD.- ANTIPCSK9 should be added- CHD is not a synonym of ACS: which type of patients do the authors intend to investigate? ACS, CAD, BOTH? Criteria for CHD definition should be provided- Please, specify why it was decided to restrict the analysis to patients under 85 yearss- Onset of what: ACS?- Type of outcome measures- Define MMP and refer to them under introduction and discussion- Discussion-statin have an anti-inflammatory effect: justify with some references-6% statin rule: please, explain - Intense anti-inflammatory treatment after PCI is also an important intervention: please, provide the reason of this statement considering that antiinflammatpry therapy has not yet been recommended by current guidelines and please use English language reference.- Page 18 PCSK9 and lipoprotein: please, cite an original article. - Therefore, in addition to its strong lipid lowering effect, PCSK9 inhibitors can alleviate inflammatory reactions and exert anti-
--

	inflammatory effects by reducing LDL-C levels, reducing LDL-R mediated platelet activation, and other ways. The sentence is not clear and apparently contrasts with what written before. Please, specify and discuss. -Bempedoic acid improves other blood lipid : please, specify, provide refs.
--	---

REVIEWER	Sezer, Murat International Hospital Istanbul
REVIEW RETURNED	04-May-2023

GENERAL COMMENTS	The authors stated that they will investigate the effect of PCSK9 inhibitors on coronary microcirculation, inflammation and cardiac function in patients with CAD after PCI. Introduction 1. Please consider organizing the introduction in a way that presents the background and rationale for this literature investigation. The title of the paper is 'Effects of PCSK9 inhibitors on coronary microcirculation, inflammation, and cardiac function in patients with CHD after PCI: A protocol for systematic review and meta-analysis.' My recommendation is to use subheadings to build an understanding for the reader than highlight the question (why you need to conduct this investigation):  • Providing a short background on post PCI microvascular dysfunction and explaining its drivers. (e.g., Coronary Microvascular Injury in Reperfused Acute Myocardial Infarction: A View From an Integrative Perspective DOI: 10.1161/JAHA.118.009949) • Rationale (Mechanisms of action by which PCSK9 inhibitors expected to affect coronary microcirculation should be discussed. Why is it worth investigating the impact of PCSK9 inhibitors on post PCI microvascular damage? Inflammation, lipid lowering action, other possible mechanisms). The possible link authors created between hyperlipidemia and coronary microcirculation should be discussed. • Stent stenosis is seem to be over discussed, at least more predominantly compared to microcirculation. The opposite would be more compatible with the title. It would be better to set a boundary to the variables on which PCSK9 inhibitors may have an effect, such as only coronary microcirculation and inflammation. Objectives 1. Primary objectives of the study were not compatible with the aim and the title of the study. These can only be some of the variables mentioned in the collected studies. The authors must use the measures of coronary microvascular damage to conduct an investigation compatible with the title and the purpose of ' Thus we propose this protocol to systematically evaluate the effects of PCSK9 inhibitors on coronary microcirculation, inflammation, and cardiac function in patients with CHD after PCI, and to provide high-quality evidence-based evidence for the clinical application. Please consider revising.. Methods
--

	1. Most importantly, authors should clearly define their criteria in the assessment of coronary microcirculation. In other words, what types of methodology used in studies in interrogation of coronary microcirculation after PCI will be chosen to be included? What does coronary blood flow classification mean?? What does it stand for? Please explain clearly .. Discussion 1. There is nothing about coronary microcirculation or microvascular damage. This is central in this investigation according to the title and the proposed purpose. In general the organization of the paper, introduction, analysis and very importantly discussion should be clearly more focused on 'Coronary Microcirculation'. Other 1. Throughout the manuscript the terms 'surgery' and 'PCI' are repeatedly interchangeably used. Surgery almost always refer to CABG in the context of coronary artery disease. This should be corrected in entire text. Minor 1. Introduction Line 68: Use of the term 'dredging' to define what PCI does to coronaries sounds inappropriate to me. 2. There are multiple inadequate quality sentences. Professional language editing or proofreading by a native speaker is recommended. ('Considering the cardioprotective effect of PCSK9 inhibitors, and there is currently no literature evaluate its effects after PCI. ' etc.)
--	---

VERSION 1 – AUTHOR RESPONSE

Reviewer 1

- The quality of writing which definitely requires native speaker revision.

Response: Thanks to the editor for your revisions, which we have accepted and made changes accordingly.

Methods

- Table 1 and search strategy: it is not clear where the authors refer to CHD.

Response: Thanks to the editor for your revisions, which we have accepted and made changes accordingly. We have added search terms for CHD, as follows: (Coronary artery disease) OR (Arteriosclerotic heart disease) OR (CAD) OR (CHD) OR (acute coronary syndrome) OR (chronic coronary syndrome) OR (ACS) OR (CCS).

- ANTIPCSK9 should be added.

Response: Thanks to the editor for your revisions, which we have accepted and made changes accordingly. We have added search terms for ANTIPCSK9, as follows: (proprotein convertase subtilisin/kexin type 9 monoclonal antibody) OR (PCSK9 inhibitors) OR (evolocumab) OR (alirocumab) OR (Praluent) OR (Repatha).

- CHD is not a synonym of ACS: which type of patients do the authors intend to investigate? ACS, CAD, BOTH? Criteria for CHD definition should be provided.

Response: Thanks to the editor for your revisions, which we have accepted and made changes accordingly. Definition criteria of CHD have been provided: CHD includes acute coronary syndrome(ACS) and chronic coronary syndrome(CCS). We intend to investigate both ACS and CCS.

- Please, specify why it was decided to restrict the analysis to patients under 85 years.

Response: Thanks to the editor for your revisions, which we have accepted and made changes accordingly. After reviewing the extensive literature, we considered the decision to modify the age limit of ≤ 85 years to < 65 years. Although there is no age limit for PCI in the CHD population, studies have shown that patients ≥ 75 years have higher rates of comorbid hypertension, peripheral vascular disease, heart failure, chronic obstructive pulmonary disease, and stroke compared with patients < 65 years. The in-hospital mortality rate was $< 0.5\%$ for patients < 65 years, whereas the mortality rate for elderly patients ≥ 75 years ranged from 2.2 to 4.0%. Given the reality of high comorbidity and high hospitalization mortality in elderly patients with CHD ≥ 75 years of age, we decided to limit the age of the study population to less than 65 years.

- Onset of what: ACS?

Response: Thanks to the editor for your revisions, which we have accepted and made changes accordingly. This part where we really want to express is the patients within 72 hours after PCI.

- Type of outcome measures.

Response: Thanks to the editor for your revisions, which we have accepted and made changes accordingly. After consideration, we decided to categorize the study's outcome indicators into three categories: primary efficacy indicators, secondary efficacy indicators, and safety indicators. After consideration, we decided to categorize the study's outcome indicators into three categories: main efficacy indicators, secondary efficacy indicators, and safety indicators.

- Define MMP and refer to them under introduction and discussion.

Response: Thanks to the editor for your revisions, which we have accepted and added explanations in both the introduction and discussion sections.

Discussion

-statin have an anti-inflammatory effect: justify with some references.

Response: Thanks to the editor for your revisions, we have accepted and added the appropriate references, eg. 44. Satny M, Hubacek JA, Vrablik M. Statins and Inflammation. *Curr Atheroscler Rep* 2021;23(12):80. doi: 10.1007/s11883-021-00977-6 [published Online First: 20211201]

- 6% statin rule: please, explain.

Response: Thanks to the editor for your revisions, which we have accepted and added explanations in the discussion sections. The "6% effect" of statins refers to the fact that doubling the dose of the drug can only result in a further 6% reduction in density lipoprotein cholesterol levels, with a significantly higher incidence of adverse effects such as rhabdomyolysis and hepatic dysfunction.

- Intense anti-inflammatory treatment after PCI is also an important intervention: please, provide the reason of this statement considering that anti-inflammatory therapy has not yet been recommended by current guidelines and please use English language reference.

Response: Thanks to the editor for your revisions, which we have accepted and added explanations in the discussion sections. After reviewing the literature, we found that the statement "Intense anti-inflammatory treatment after PCI is also an important intervention" is indeed not rigorous enough. The activation of platelet aggregation and exacerbation of inflammation after PCI has been confirmed by relevant studies, and therefore effective anti-inflammatory treatment after PCI is beneficial in improving coronary blood flow and reducing the risk of adverse cardiovascular events.

- Page 18 PCSK9 and lipoprotein: please, cite an original article.

Response: Thanks to the editor for your revisions, which we have accepted and added explanations in the discussion sections. Integrating the revisions made by the editors, we have revised the “discussion” section as a whole, removing the lipoprotein section and providing a detailed introduction to PCSK9.

- Therefore, in addition to its strong lipid lowering effect, PCSK9 inhibitors can alleviate inflammatory reactions and exert anti-inflammatory effects by reducing LDL-C levels, reducing LDL-R mediated platelet activation, and other ways. The sentence is not clear and apparently contrasts with what written before. Please, specify and discuss.

Response: Thanks to the editor for your revisions, which we have accepted and added explanations in the discussion sections. After reviewing the relevant literature we have modified the formulation of this section, as detailed in the discussion section. In addition, integrating the revisions made by the editors, we have revised the “discussion” section as a whole.

- Bempedoic acid improves other blood lipid : please, specify, provide refs.

Response: Thanks to the editor for your revisions. Integrating the revisions made by the editors, we have revised the “discussion” section as a whole, and removing the content on Bempedoic acid.

Reviewer 2

Introduction

1. Please consider organizing the introduction in a way that presents the background and rationale for this literature investigation. The title of the paper is ‘Effects of PCSK9 inhibitors on coronary microcirculation, inflammation, and cardiac function in patients with CHD after PCI: A protocol for systematic review and meta-analysis.’ My recommendation is to use subheadings to build an understanding for the reader than highlight the question (why you need to conduct this investigation):

Response: Thanks to the editor for your revisions, which we have accepted and added explanations in the introduction sections.

- Providing a short background on post PCI microvascular dysfunction and explaining its drivers. (e.g. Coronary Microvascular Injury in Reperfused Acute Myocardial Infarction: A View From an Integrative Perspective DOI: 10.1161/JAHA.118.009949)

Response: Thanks to the editor for your revisions, which we have accepted and added corrections in the introduction sections. Invasive procedures such as balloon dilation during PCI can damage vascular endothelium, induce an inflammation in endothelial cells, and promote micro-thrombosis, leading to coronary microcirculatory disorders(CMD) and myocardial ischemia-reperfusion injury(MIRI), which significantly increase the risk of major adverse cardiovascular events(MACEs) in patients during the perioperative period.

- Rationale (Mechanisms of action by which PCSK9 inhibitors expected to affect coronary microcirculation should be discussed. Why is it worth investigating the impact of PCSK9 inhibitors on post PCI microvascular damage? Inflammation, lipid lowering action, other possible mechanisms). The possible link authors created between hyperlipidemia and coronary microcirculation should be discussed.

Response: Thanks to the editor for your revisions, which we have accepted and added corrections in the introduction section.

- Stent stenosis is seem to be over discussed, at least more predominantly compared to microcirculation. The opposite would be more compatible with the title. It would be better to set a boundary to the variables on which PCSK9 inhibitors may have an effect, such as only coronary microcirculation and inflammation.

Response: Thanks to the editor for your revisions, which we have accepted and added corrections in the introduction section.

Objectives

1. Primary objectives of the study were not compatible with the aim and the title of the study. These can only be some of the variables mentioned in the collected studies. The authors must use the measures of coronary microvascular damage to conduct an investigation compatible with the title and the purpose of 'Thus we propose this protocol to systematically evaluate the effects of PCSK9 inhibitors on coronary microcirculation, inflammation, and cardiac function in patients with CHD after PCI, and to provide high-quality evidence-based evidence for the clinical application. Please consider revising.

Response: Thanks to the editor for your revisions, which we have accepted and added corrections in the objectives section. After reviewing the literature we decided to categorize objectives into three groups: Main efficacy indicators, secondary efficacy indicators and safety indicators. The main efficacy indicators included TIMI flow classification of coronary arteries, myocardial injury markers(cTn, CK-MB, Myo), NT-pro BNP and D-dimer before and after PCI. Secondary efficacy indicators: Serum inflammation markers (CRP/hs-CRP, IL-6, TNF- α), MMP-9, cardiac function indicators(LVEF, FS), blood lipid indicators(LDL-C, TC, TG, HDL-C). Safety indicators: The incidence of postoperative complications [puncture point hematoma, gingival bleeding, hematochezia, urinary blood, etc], MIRI[reperfusion arrhythmia(RA), myocardial stunning, no-reflow and lethal reperfusion injury, etc], MACEs(incidence of recurrent myocardial infarction, heart failure, cardiogenic shock, and cardiogenic death within six months after PCI).

Methods

1. Most importantly, authors should clearly define their criteria in the assessment of coronary microcirculation. In other words, what types of methodology used in studies in interrogation of coronary microcirculation after PCI will be chosen to be included? What does coronary blood flow classification mean?? What does it stand for? Please explain clearly.

Response: Thanks to the editor for your revisions, which we have accepted and added explanations in the methods section. We chose TIMI flow classification to assess the coronary microcirculation in CHD after PCI. It was divided into 4 grades: grade 0 refers to the absence of contrast medium filling at the distal end of the vessel, suggesting that there is no blood perfusion at the distal end, indicating that the vessel may have a completely occlusive lesion; grade 1 refers to the stenotic part of the vessel with partial visualization of the contrast medium, but the contrast medium can not reach the distal end of the vessel well, suggesting that the stenosis of the vessel is more severe, with a lesion close to occlusion; grade 2 refers to the ability of the contrast medium to fill the vessel and make the vessel visualized, but the speed of the development is slower than that of normal vessels, suggesting that there is some stenosis or lesion in the coronary arteries. Grade 3 means that the contrast medium can fill the vessel quickly and completely, indicating that the blood flow status is normal.

Discussion

1. There is nothing about coronary microcirculation or microvascular damage. This is central in this investigation according to the title and the proposed purpose. In general the organization of the paper, introduction, analysis and very importantly discussion should be clearly more focused on "Coronary Microcirculation".

Response: Thanks to the editor for your revisions, which we have accepted and added explanations in the introduction and discussion section.

Other

1. Throughout the manuscript the terms 'surgery' and 'PCI' are repeatedly interchangeably used. Surgery almost always refer to CABG in the context of coronary artery disease. This should be corrected in entire text.

Response: Thanks to the editor for your revisions, which we have accepted and made corrections.

Minor

1. Introduction Line 68: Use of the term 'dredging' to define what PCI does to coronaries sounds inappropriate to me.

Response: Thanks to the editor for your revisions, which we have accepted and made corrections.

2. There are multiple inadequate quality sentences. Professional language editing or proofreading by a native speaker is recommended. ('Considering the cardioprotective effect of PCSK9 inhibitors, and there is currently no literature evaluate its effects after PCI.'etc.)

Response: Thanks to the editor for your revisions, which we have accepted and made corrections.